# Global Quality Management System (G-QMS) in Systems of Systems (SoS)—Aspects of Definition, Structure and Model

**Noga Agmon, Sigal Kordova *** and **Shraga Shoval**

Industrial Engineering and Management Department, Faculty of Engineering, Ariel University,
Ariel 4077625, Israel; agmonn@ariel.ac.il (N.A.); shraga@ariel.ac.il (S.S.)
* Correspondence: sigalko@ariel.ac.il

**Abstract:** The present study is innovative in its integration of several novel and quickly developing disciplines of *QMS*, *SoS*, *Globalization* and Systems approaches including *Systems Thinking*. We inaugurate *G-QMS in global SoS organizations* as a new field of research. This is an exploratory study that uses the *Grounded Theory* combined with an *analytical review* and *professional experience* to provide a framework for identifying of new key variables in the multidimensional environment of global management. The purpose of this study is to create a theoretical foundation for this field, and introduce logical deductions regarding G-QMS in global SoS organizations that can be used as foundational principles for defining and modeling of G-QMS. The methodology of the study includes a paradigm that combines *analytical review*, which integrates the four main disciplines into a *structured qualitative study* based on semi-structured interviews, and used Grounded Theory. The findings show that G-QMS is a necessary condition for these organizations, while the management of G-QMS is inseparable from the management of the SoS. The final results reveal *18 aspects* to be considered in any definition determined for G-QMS in global SoS organizations, and any model to be developed. From these, *8 base anchors* for the model were analyzed and mapped, as well as its main factors. In conclusions, each of these base anchors makes its own contribution to any further development in this area. However, considering them all together creates an *initial model* of *G-QMS in global SoS organizations*.

**Keywords:** Quality Management System (QMS); Global Quality Management System (G-QMS); System of Systems (SoS); globalization; international organizations; global management; systems theory; systems thinking

## 1. Introduction

### 1.1. Rationale

The current research integrates, for the first-time, the relatively new and rapidly evolving disciplines of *Quality Management System (QMS)*, *System of Systems (SoS)*, *Globalization* and Systems approaches such as *Systems Thinking*, by defining a field of research for *Global Quality Management System (G-QMS) in global SoS organizations*. This is a very relevant, innovative field of academic research that is also applicable to business organizations. The current study presents a QMS that includes references to aspects of *global organizations* and also to those relevant to *SoS organizations* that are missing in the existing international standards for QMS. In order to support these missing aspects, the *Process Approach* that underlies the ISO international standards for QMS is expanded by introducing *Systems approaches*, in particular, *Systems Thinking*. Figure 1 outlines the field of the research by illustrating the disciplines it incorporates.

*G-QMS in global SoS organizations* is not yet defined. The current literature lacks not only a definition but also defined structures and standards, and there is significant motivation to advance and develop beyond this situation. The current paper presents an exploratory study that uses *Grounded Theory* combined with *analytical review* and *professional*

*experience* to create the theoretical foundation for G-QMS in global SoS organizations. It creates a framework of knowledge for this field, which is based on an analytical review that integrates the four main disciplines which are involved in this field of research—*QMS*, *Globalization*, *SoS* and *Systems Thinking*, and on the vast professional experience of two decades in leading of QMS in global and advanced technology companies. It is further supported by a preliminary study utilizing the Grounded Theory.

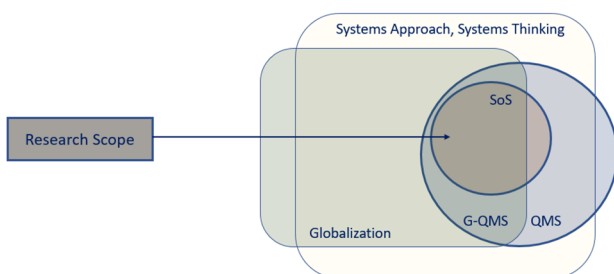

**Figure 1.** A view of the overlap among the disciplines incorporated in this study.

*1.2. Background and Relevance*

Quality Management in SoS organizations are characterized by very complex products and organizational structures. Since SoS integrate multiple systems and technologies, the corresponding organizational systems are characterized by great complexity that includes sub-organizational substructures. These organizations usually include various units located at multiple sites, and usually have a global geographical layout, meaning that they are global organizational systems. Such organizations require an efficient and effective organizational system that benefits from operating the structural and global system in a multifaceted, demanding and ever-changing environment, including ever-changing technology and multiple and diverse environments. Their QMS must support both current ever-changing complexity and future adaptations, meet customer demands and regulatory requirements, while also giving the organization added value by being a superlative, tailor-made QMS that is exceptionally relevant, effective and efficient. Gorod et al. [1] review the current literature on SoS, and found that is still in an embryonic state in terms of identifying an effective methodology for achieving the objectives of SoS, and generally defining SoS organizational systems. Furthermore, SoS has grown in importance and correspondingly, there is increasing interest in its concepts and strategies. In this context, there is the motivation to define the relevant QMS for these organizations, which is also a global one.

A QMS is based on compliance with requirements of international standards, that set a global standard. The main and most commonplace one is ISO 9001, which embodies the best practices of quality management and quality certification, and suits a wide range of organizations, and all the other similar standards are based on it. The international standards refer to the QMS of *organizations*. They, refer to all the sizes and types of organizations, from local and single-site organizations to multi-site and global operations (large conglomerates), from those that produce a relatively simple product to those whose products are complex/composite systems or SoS. However, in actual practice, they lack the needed references which are relevant and necessary in highly complex organizations, and organizations with a global geographical distribution—*global SoS organizations*.

A QMS is based on the main principle of these international standards, the Process Approach. When dealing with it, the Business Process Orientation (BPO) approach and Process Maturity are also incorporated. Accordingly, the corresponding Process Approach that underlies the QMS must be scalable for dealing with complex organizations and those having a global organizational system, particularly complex/composite systems and SoS organizations.

QMS is based on compliance with an entire set of requirements, e.g., ISO 9001 [2] and similar. Assurance or evidence of compliance with these requirements make them

eligible for Certification according to the standard, and to maintain certification over time. That is, as long as the standard's certification is maintained, the evidence of compliance with all QMS requirements is maintained. Nonetheless, standard certification is based on a 0/1 model, either there is compliance (certification) or there is not. This dichotomous model for the wide range of requirements required from a QMS in fact misses (or lacks) reference to other Process Approaches that express the dynamics of processes. As a result, the entire set of requirements (which is broad and comprehensive, but also subject to applied interpretation) is perceived in terms of the first level, and first level only. This set of requirements does not provide a sufficient organizational solution for SoS organizations, especially international SoS. In this context, wider, more comprehensive requirements have already been formulated in the form of guidelines that do not require certification (for example, ISO 9004 [3]). However, even ISO 9004, similar to ISO 9001, does not address the aspects essential to dealing with global SoS organizations.

SoS organizations, as well as other global organizations characterized by high complexity, and very complex QMS, actually miss (or lack) a reference to *systemic approaches*. In this regard, the comprehensive requirements—guidelines of ISO 9004—can be extended in terms of System Maturity to these kinds of organizations. This systems approach is considered alongside other Systems approaches such as Systems Thinking—an interdisciplinary conceptual framework that is reviewed and described in the Literature Review.

Global Quality Management System (G-QMS) is a new term that has not received sufficiently attention in the research arena, nor is it clearly expressed in the quality management standards (although they are international standards). The somewhat flexible and ambiguous definition of the Process concept in the ISO 9001 and ISO 9004 does not address the level of process complexity of global organizations [4], including their systemic complexity. Furthermore, there is still no global quality management strategy, nor is there a set of requirements or guidelines for it. The Process Approach on which the international quality management standards are founded has the capacity to be expanded. In this broadening Systems approaches shall be considered. Extending the Process Approach underlying the QMS standards is necessary and consistent line with the perspectives and principles of systems theories as well as those of the Systems Thinking. When dealing with organizational systems and QMSs in SoS organizations, systematic approaches should be incorporated. The perspectives and principles (and possibly even the tools) of the System Approaches might be integrated in the structures and characteristics of SoS. According to the SEBoK [5], SoS is a relatively new area, and therefore only limited attention has been given how Systems Thinking might be extended to the issues particular to SoS.

This research field is essentially contemporary and relevant. QMS is a developing field that is increasingly required by organizations, with both internal and external motivations. The internal motivation responds to customer requirements for increased product quality, together with intolerance to faults and delays. This is especially the case in more complex the organization is (such as SoS), where it is increasingly necessary to deal with product quality and delays in customer delivery. Externally, the motivation responds to regulatory requirements, which constantly evolve and "deepen their grip" on both the product and the organizational processes for its realization.

Each of the disciplines encompassed by this field of research is relatively new and rapidly evolving. International standards for QMS are developed and updated once a decade, and organizations adapt themselves to the updated requirements accordingly. Nevertheless, international standards of QMS are focused on the *organization* as a classic entity and lack the required references to the organizational framework, attributes and characteristics of the SoS. That is, their relevance to the reality in which SoS organizations grow and become established, and with it academic research on SoS, should be evolving. Likewise, international standards of QMS lack the reference to the globalization aspects which are relevant in more and more organizations in today's interconnected, global world, particularly SoS organizations. Moreover, the academic literature on global-QMS is also in an embryonic state. Systems Thinking is also a new, evolving discipline that

still lacks a commonly-accepted definition or understanding [6]. However, it offers a perspective and tools that are pertinent and can be supported when integrating QMS in global SoS organizations. Systems Thinking can provide a substantial power and value, which are the basic infrastructure for developing frameworks for elements such as structure, motion, dynamics, interrelationships and interactions, including interrelationships with the environment.

*1.3. Purpose*

The purpose of the study is to create a theoretical foundation for this field and introduce logical deductions regarding G-QMS in global SoS organizations that will be used as foundational principles for defining and modelling G-QMS. This exploratory study examines a modeling framework for G-QMS in SoS and provides a starting point for further research on developing G-QMS in global SoS organizations. This G-QMS should be based on QMS requirements and guidelines as proven by the international standards as well as those adopted by most organizations, and also consider Systems approaches that can be incorporated in the suggested structure of G-QMS. Furthermore, G-QMS which shall operate as an integral part of these kinds of organizations, and one that would be the appropriate and relevant.

## 2. Literature Review

The review includes the disciplines related in the field of research in a way that will facilitate integrating the separate disciplines, which is essentially the basis of the research work. It also reviews disciplines that are relatively new fields, which have mostly evolved over the last two decades.

*2.1. What Is QMS and Its Motivation in Organization?*

The field of Quality Management in organizations has been developed over the past decades and is framed as a Quality Management System (QMS), a system that strives to improve the overall performance of an organization and provide a sound basis for sustainable development initiatives [2]. A QMS helps coordinate and direct an organization's activities to meet customer and regulatory requirements and improve its effectiveness and efficiency on a continuous basis. International standards for QMS, by which the industry works, provide its definition and scope. They specify a set of requirements that an organization needs to assimilate and implement in order to establish and maintain a QMS in the organization. ISO—the International Organization for Standardization—is a worldwide federation of national standards bodies. The most widely used standard and for a large range of organizations is ISO 9001, which lists all requirements for a QMS.

The QMS is based on the Process Approach, which is explicitly mentioned as the main principle of these standards: "This International Standard is based on process approach to quality management" [2] and likewise in other standards. According to the Process Approach, consistent and predictable results are achieved more effectively and efficiently when activities are understood and managed as interrelated processes that function as a coherent system. Academic literature also refers to the Process Approach as used in the international standards for QMS as BPO, which is a concept that has been adopted by companies worldwide. Companies that wish to improve their performance and stay competitive are introducing and adopting a process view of business in order to enhance their overall performance [7,8]. The QMS consists of interrelated processes, and understanding how results are produced by this system enables an organization to optimize the system and its performance. The Process Approach/BPO integrates processes into a complete *system* for achieving strategic and operational objectives.

Further to the principles and requirements of ISO 9001, ISO 9004 expands and deepens the QMS, which "provides for organizations to achieve sustained success in a complex, demanding and ever-changing environment, with reference to the quality management principles described in ISO 9001" [3]. Likewise, while ISO 9001 focuses on providing

confidence in an organization's products and services, ISO 9004 addresses the systematic improvement of the organization's overall performance [3].

### 2.2. QMS in Complex and Global Organizations

The current research deals with QMS in organizations that develop and operate *complex systems products*, and in particular SoS. These organizations have a high level of complexity that emerges from several primary factors that are each complex in their own right: (1) Complex system product is usually multi-disciplinary product involving a relativity high number of different actors from many disciplines, having diverse relationships and numerous integrated systems. (2) An organizational system that is characterized by a multiplicity of processes, and also processes which are complex, including multiplicity of interfaces and interactions. (3) The current reality of complex, demanding and ever-changing environments and technologies. Additionally, the current research focuses on the QMS of the complex organizations which are also global, that have global organizational system and expand into international markets using various strategic options and international operational systems, with global customers and global suppliers. Such organizations have complicated interfaces with both suppliers and customers, and also internally according to their international organizational layout. Furthermore, they operate in a complex, demanding, multi-site environment that changes continually.

The discipline of quality management, although subject to regulatory and standard requirements, is not a dichotomous field where one answer is correct and another wrong. Standardization and regulation are legal realms, and therefore different ways can be found to achieve or comply with the same requirements and guidelines. In fact, the international standards for QMS leave a wide range for applied interpretations of their requirements and guidelines, thereby revealing some flexibility in the QMS structure. Thus, when dealing with complex and global organizations, this space can potentially be utilized to promote a model for QMS structure that will be suitable, relevant and efficient for supporting this kind of organizations.

Businesses in the 21st century operate in an increasingly complex global environment. The global reality and its defining characteristics highlight the need to change how quality management is perceived and implemented, given the increasing complexity and multiplicity of mutual relationships between production, product, services and network processes [9]. The global economy and its characteristics have had a profound impact on the development of the concept of quality worldwide. Likewise, issues related to quality management receive added weight in light of the need to function at a very high level of complexity, which demands intricate administrative and managerial strategies [10]. Globalization provides organizations with many opportunities; however, it also presents management systems with complex challenges. With respect to the organizational QMS, beyond the classical issues associated with local operations, multinational organizations must also cope with challenges presented by the decentralization of the organizational functions. The global setting creates particular difficulties for quality assurance and control managers at different levels, and therefore there is an incentive to define it in regard to the organizational QMS and lay the foundation of a global-QMS, that will be relevant and effective for complex system product organizations which are also global.

### 2.3. Global Quality Management System (G-QMS)

As can be concluded, there is great potential in the subject of G-QMS and research that focuses on complex system, global organizations and, especially SoS organizations. Although QMS challenges can be naturally expanded to G-QMS, dealing with G-QMS in global organizations with complex, large-scale systems, requires relating to additional aspects specific to organization of this kind and their characteristics.

A review of the literature on the topic of G-QMS indicates that there is a need for a solid G-QMS philosophy, with emphasis on intra-organizational coordination and process management, but this is remained undefined. According to Kim and Chang [11], Total

Quality Management (TQM), which was the conventional term for QMS at the time, addresses a single organizational level, and questions remain open as the discussion expands to the global level. They extend the TQM concept to what they consider to be the next evolutionary stage, Global Quality Management (GQM). In comparing TQM to GQM, the strategic concepts behind TQM are maintained, but GQM enlarges the scope to encompass the concerns of multiple functions across multiple countries, and therefore the subsequent increased complexity of GQM. This requires moving beyond TQM to develop a quality concept that reflects the nature of global corporations and their markets. They called this a basic concept "first seeds" for GQM and noted that further development of the term GQM is needed. Despite the growing use of the term, there is still no agreed definition for GQM, nor is there a universal agreement upon the concept itself [12]. QMS in turn requires systematic leadership innovation. The bigger an organization, the more difficult it is to ensure the unity of goals manage its knowledge and govern changes. Formation of a QMS for a complex economic entity not only shapes the environment for the effective, innovative development of a parent company and its separate business units, but also represents an independent organizational innovation that is able to boost the diffusion of organizational changes and launch the expansion of innovations [13].

The findings reported by Bashan and Notea [4] suggest multiple conflicts and a lack of clarity regarding the integrative management of the various Quality Systems within a corporate group, highlighting the need to plan relevant integration mechanisms for Quality functions. Such integration mechanisms are meant to ensure that the different Quality Systems within a multinational company will be mutually connected and function as part of a single organizational entity, in order to reduce failures and ensure Quality across global processes. According to Barabasi and Frangos [14], who use the term "network organizations" to describe the organizational structure of complex and global organizations, network organizations are formed as part of the expansion process of a company. This creates synergy between organizations, or within a meta-organization, moving from a tree structure to a multi-dimensional network structure. This is one of the most significant changes in the area of complex systems; it results from global competition and raises the question of the level of globalization of the quality system.

*2.4. SoS Organizations*

Gorod et al. [1] review the literature on SoS and show that although SoS is a relativity new term, there has been significant development in research and experimental applications in the field, mostly during the last two decades. However, the relevant literature shows that research on SoS is still embryonic, both in terms of identifying an effective methodology to achieve the objectives of SoS and concerning the relevant organizational systems. SoS evolved from the earlier System discipline, which has been studied and developed throughout the latter half of the 20th century. However, a process of rapid global acceleration, especially in the military sector, continued and this made expanding developments in engineering to the next level essential. The objective was to address "shortcomings in the ability to deal with difficulties generated by increasingly complex and interrelated system of systems" [15]. The need for a discipline focused on engineering multiple, integrated, complex systems, led to the emergence and evolution of SoS as a discipline, as evident in the literature since the early 90s. Interest continues to accelerate even today.

Much of the literature deals with the need to define SoS and provides a variety of initial definitions [1,15–17], but there is still no agreed definition of SoS [18]. SoS moves the focus from single systems to multiple, integrated complex systems. However, the basic principles of complex systems can be applied to SoS, making it is imperative to use complex systems as a foundation for the research in the field of SoS [1]. The transition to the accepted modern term SoS is introduced in the works of [19–22]. A recent definition can be found in ISO/IEC/IEEE 21839:2019 [23], which also provides a definition for Constituent Systems. SEBok [5] also uses the term Constituent Systems when describing SoS.

Azarnoush et al. [24] emphasize that there is an increasing interest in exploiting synergy between these independent systems to achieve the desired overall system performance. In 2007, the Department of Defense (DoD) published their System of Systems Engineering Guide: Considerations for Systems Engineering in a System of Systems Environment [25], which describes the characteristics of SoS environments and identifies complexities of the SoS.

While the literature in SoS is expanding rapidly, there is no established body of knowledge, and nor management framework that guides our understanding of these complex systems [1]. Gorod et al. [1] presented a SoS management framework which brings together a leading approach to describing SoS (i.e., characterization) and one of their fundamental traits (i.e., networks). Likewise, work is needed to identify and articulate the cross-cutting principles that apply to SoS in general, and to develop working examples of the application of these principles [5]. While System of Systems Engineering (SoSE) is not a new discipline, this is an opportunity for the Systems Engineering (SE) community to define the complex systems of the 21st century [26]. While SE is a fairly established field, SoSE represents a challenge for the present systems engineers on a global level [5].

### 2.5. Systems Thinking and the Systemic Approach

In order to promote a G-QMS in SoS organizations, it is necessary to draw inspiration from and integrate aspects of Systems Thinking and the systemic approach. Systems Thinking is also an emerging domain with many different views regarding its definition, rather than one that is precise and widely-accepted. Monat and Gannon [6] conclude, based on a literature review, that Systems Thinking is a perspective which uses a language and a set of tools. Specifically, it is the opposite of *linear thinking*, and focuses on the *relationships* among components of a system, as opposed to the components themselves. It is *holistic* (integrative) thinking instead of analytic (dissected) thinking. The following literature review focuses on Systems Thinking in the domain of organizational systems, and the motivation for its inclusion in this study. It is limited to matters of scope and definition without presenting examples and applications, although they are widely included in the literature.

Systems Thinking is founded on General Systems Theory [27] and has been applied to a wide range of fields and disciplines. It has great power for solving complex problems that cannot be solved using conventional, reductionist thinking [28–30], and can be used to explain dynamic, non-linear and complex organizations and environments. Checkland [31] draws a distinction between "hard" and "soft" Systems Thinking. He states, "Systems Thinking, makes conscious use of the particular concept of *wholeness* captured in the word 'system' to order our thoughts", and "Systems Thinking implies thinking about the world outside ourselves". Senge [29] provides a generic definition: "Systems thinking is a discipline for seeing *wholes*. It is a *framework* for seeing interrelationships rather than things, for seeing patterns of change rather than 'snapshots'". Senge's book presents the background and theory of Systems Thinking, and is pivotal because it applies Systems Thinking to management in organizations. Afterward Anderson and Johnson [32] applied Systems Thinking to practice by defining it as a set of tools, a framework for looking at issues, and a language. Richmond [33] "sees the wholes" and considers Systems Thinking the art and science of making reliable inferences about behavior by developing an increasingly deep understanding of the underlying structure. Systems Thinking can provide a great deal of power and value when dealing with complex and everchanging organizations. Within organizational systems, it deals with elements including *organized complexity*, *system dynamics*, *self-organization* and *structure based on processes* and *relationships*, interconnections as well as emergence. All of these are relevant in both G-QMS and SoS organizations [34]. Thinking systemically also requires several shifts in perception, which lead in turn to different ways to arrange an organizational system [35,36].

Systems Thinking is founded on a *holistic* perspective, thus it does not try to break systems down into components in order to understand them; rather it focuses on how the

components *act together* in networks of interaction. Consequently, the only way to fully understand a system is to understand its components as they relate to the *whole*. That is to say, "the whole is greater than the sum of its parts" or "seeing the big picture". Indeed, tackling a problem in its entirety often provides a much more effective solution. Bashan and Kordova [37] introduce a system view of globalization and quality management and highlight the concept of Systems Thinking in the global world. Since it is an interdisciplinary conceptual framework, Systems Thinking can be used as a tool in complex systems organization such as SoS and global organizations. When systems perspective is integrated into the working environment of global organizations, it can form part of its infrastructure and facilitate use of terminology and tools in an environment which is increasingly characterized by the ever-changing complexity often associated with global organizations.

The high level of complexity that characterizes the competitive global environment generates the need to adopt a systems perspective for analyzing how a multinational company develops, and the effect this development has on global quality functioning performance. Specifically, the *systems analysis approach*, which is based on open and complex systems theories, is suitable. In their discussion about the ability of ISO 9001 and ISO 9004 meet the needs of a global quality system, Bashan and Armon [12] recommend adopting an Open System Approach that refers on cross-organizational processes, and the complex of interfaces and interactions that exist between them and their external environment. The classical Process Approach does not provide a framework for the dynamic and ever-changing processes occurring during global company expansion. Rather, global reality requires a different process structure. The literature review shows that Globalization is a new phenomenon, and insights into its complexity and management methods, including management systems, particularly QMS, are still immature. The need for systemic analysis that addresses this complexity motivations the adoption of systemic analysis tools such as Systems Thinking and Systems approaches, which can be integrated into quality management processes at the global level and contribute to a better understanding of their complexities and diverse dimensions.

In summary, Systems Thinking provides perspective and can be a powerful tool in complex and global organizational systems. It is relevant to and can be supportive integrating QMS in global SoS organizations, but its use in most organizations remains insufficiently developed. The current revision of the SEBoK devotes a chapter to Systems Thinking. However, as a compendium of literature articles on Systems Thinking concepts, principles, and patterns, the chapter is quite vague and does not appear to integrate the disparate articles into a cohesive whole [5].

*2.6. Two Additional Aspects When Developing G-QMS in SoS Organizations*

(a) System Approach and System Maturity in G-QMS

Schematically, the System Approach is an expansion of Process Approach for organizations of relativity high complexity, typically global and SoS organizations. Similarly, System Maturity is an expansion of Process Maturity, a model for assessing and/or guiding best practice improvements in organizational maturity and process capability, expressed on a lifecycle level. Process Maturity is used as an indication of how close a process is to being complete and how capable of continual improvement through qualitative measures and feedback. When dealing with System Maturity, an integrated methodology is introduced to assess system maturity and performance throughout the lifecycle of a system, from concept development, through design, and ultimately to the operational deployment of the system. Yet the literature shows that there is no single integrated methodology for System Maturity.

ISO 9004, which provides guidelines which expand and deepen the requirements for QMS, promotes the use of System Maturity by introducing, in an appendix, a five-level maturity model. Its detailed extension to the entire set of guidelines allows the reference to Systemic Maturity, while this tool is used as a reference in the current study [3]. Moreover, other international standards for QMS such as the Sectorial Standards are in development in accordance with this trend.

(b)    Reference to the Sectoral QMS Standards

In the last two decades, there has been a trend of differentiation in the realm of quality management, according to sectors of business activity. Quality management standards that began to differentiate in the 1st decade of the 21st century already include the second revision. These international standards include AS9001—QMS requirements for Aviation, Space and Defense organizations [38], ISO 13485—for Medical Device organizations [39], ISO 22163—for Rail organizations [40] and IATF16949—for Automotive organizations [41]. This trend of the sectoral directives is contemporary and actual, with sectoral international standards expanding and deepening the applicability of QMS. Moreover, these sectors are relevant to complex systems and global organizations. This differentiation trend has evolved to provide the applicability of QMS in organizations that develop and operate complex and large-scale systems in a demanding, ever-changing reality. In addition, these sectors are characterized by a *strong customer orientation*, which is an environment where customers are involved in setting the requirements for the standard.

The current study focuses on global organizations that develop and produce complex SoS products, SoS organizations, for short. The Sectoral QMS Standards above are typical in SoS organizations and include expanding requirements, especially as regards organizational relationships that include Customer-Supplier chains. In this realm, too, the holistic definition required for a G-QMS is still lacking.

## 3. Methodology and Research Design

### 3.1. Methodology

The research paradigm combines *analytical review* and *structured qualitative study*. The analytical review deals with the four main disciplines discussed in the literature review—QMS, Globalization, SoS and Systems Thinking—and focuses on the actual and potential interrelationships and interactions among disciplines, in order to create a framework of knowledge for this field of research. This structured, qualitative study uses Grounded Theory as its theoretical and methodological framework [42,43]. Grounded Theory is a complex iterative process of asking generic questions to help guide research and generate theories by exposing patterns that emerge from data. Once data has been gathered, core concepts can be identified, and connections develop between the data and the core theoretical concepts. Grounded Theory is built on the basis of sources and findings, rather than on a theory that is already definitively established; its purpose is to add knowledge; not to confirm or refute an opinion or judgment about the phenomenon being studied. Respectively, the research structure is flexible and general, and indicates the way forward. The theory produced is grounded in reality as it is constantly constructed based on comparisons and contradictions found in the field-contributions. Therefore, the value of research is determined by its contribution to the body of existing theoretical knowledge, and description or understanding of the researched phenomenon. Since this study is a preliminary one, qualitative research is used to develop awareness and understanding of concepts, describe different aspects of reality, and contribute layers of knowledge to the theoretical body, according to the Grounded Theory methodology.

The qualitative research is based on expert interviews. The participants in the study were selected for their expertise in the disciplines of QMS, Globalization, SoS and Systems Thinking, with some interviewees having expertise in a combination of these disciplines. Professionals and executives from both academic and business environments were included.

This qualitative approach attaches importance to the meaning of things as they are perceived by the respondents, and facilitates exposing these perceptions, thereby helping researchers understand internal processes that are usually not visible to outsiders. Moreover, the researched reality is perceived as a set of interactions (between the respondents and the statements of each respondent), thus the study makes it possible to observe the entire phenomenon, and even interpret it from the perspective of the respondents (who share it and are experts in it). It is, "research **with** experts to be queried and not **about** queried experts" [43]. Qualitative research, in essence, seeks a subjective understanding or

interpretation of the phenomenon being studied, in a way that deepens the potential for learning. Therefore, in order to ensure the objectivity of the study, rules were established for how the study was conducted and analyzed, as described below, in Section 3.2.

### 3.2. Research Design

The research paradigm combined analytical, quantitative, and qualitative methods, as presented in Figure 2. The qualitative research was based on semi-structured interviews. The analytical component included both the content analysis methods of the research data and secondary sources (literature). The quantitative component complemented the analytics and scored the responses, determining the assessment scale values and the cross-content analysis.

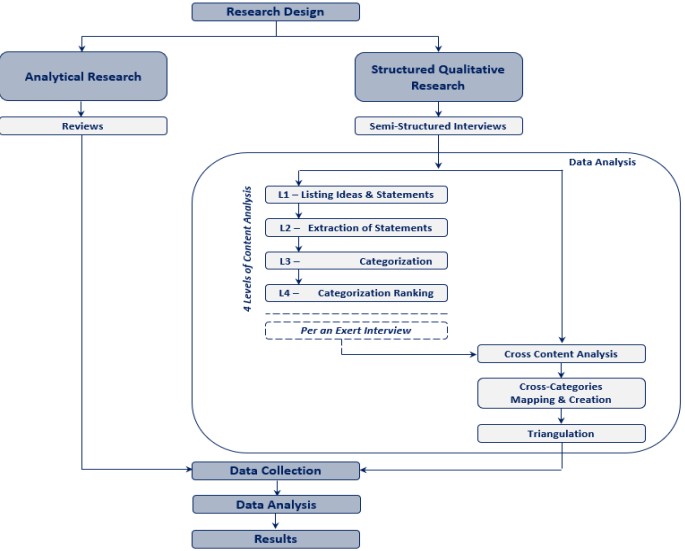

**Figure 2.** The research design.

### 3.2.1. Exploratory Study

The qualitative research was conducted as an exploratory study, involving experts who were carefully selected to represent the "state of the art" in their respective fields, as per the Research Scope depicted in Figure 1. They included experts, leaders and senior in academia, economics, and also executives in SoS organizations. Each expert was mapped according to his specific professionalism in a way to ensure full coverage of the research arena and furthermore, to aim the focus on the "Research Scope" by directing each professional "spotlight" over it, in order to of gain insight into how each one sees and relates to the "G-QMS in global SoS organizations". Figure 3 presents an overlap view combining the disciplines relevant to the "Research Scope" and expertise of the seven participants.

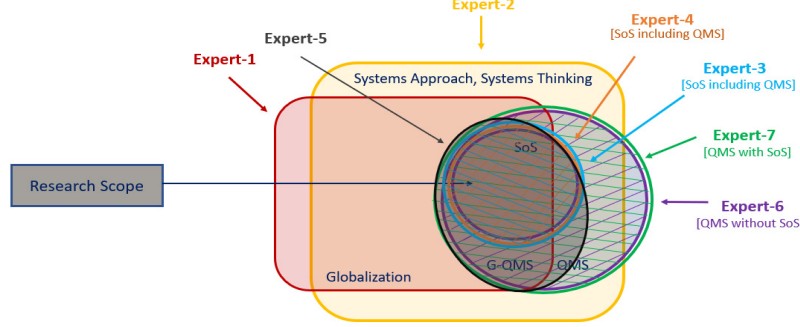

**Figure 3.** Overlap view of the disciplines included in the Research Scope, together with the participants' areas of expertise.

The semi-structured interviews were conducted with each expert in-depth manner that specified interview contents with regards to the expertise of the interviewee, and included a duration of more than one session. However, they included a set of questions tailored to the experts in a way that would allow for structured and cross analysis of them all. The interviewees were asked about definitions and concepts; the motivation and need for G-QMS in SoS global organizations; the required structure of G-QMS in global SoS organizations as regards to existing ISO standards; the required structure of G-QMS in global SoS organizations in light of their special characteristics and needs, as well as their strengths and potential weaknesses. Questions were asked regarding process approaches and methods that could contribute to the definition of the G-QMS structure in global SoS organizations, particularly the Maturity Model and Systems Thinking. Finally, the experts were asked, to look ahead and relate to G-QMS's contribution to global SoS organizations, and about the conditions needed for the effective growth of G-QMS in those organizations. The study interviews were conducted in a planned sequence during a year period.

In accordance with the qualitative research methodology, the initial research questions were broad, *conceptual questions* to investigate the processes and occurrences and arrive at the necessary clarifications. The interviewers used *open-ended questions* through which more information can be gathered and a deeper understanding gained; this contributes to learning during the research process and enhances understanding processes, interrelationships and meanings, and adds to knowledge. In order to ensure validity and trustworthiness at this phase, all the interviews were conducted in the same manner and format, subject to the same defined rules and procedures.

### 3.2.2. Data Analysis

Data analysis included the use of strategies that demonstrate different techniques for data analysis using *content analysis* including, *analytical induction*, *constant comparison*, and *counting and quantification methods* such as examining the criteria and consistency in the data collection techniques. In order to ensure validity and trustworthiness, additional measures incorporated into the content analysis, and applied in the same manner for all interviews. The content analysis process included four levels that enabled the gradual identification of categories, followed by analysis for intensity and frequency of repetitions. Each level was completed for all interviews prior to starting of the next level.

Setting the parameters: ***Quantity*** is the number of times a category appears in the statements and idea extractions. Quantity represents items that are repeated by the interviewee throughout the interview, thereby contributing to the validity of the statements. Moreover, Quantity reflects not only repetitions by the interviewees, but also words that were emphasized and the use of examples to expand and deepen an idea. The higher a category's Quantity, the more valid it is deemed. ***Strength*** is the level of importance attributed to the category, in order to differentiate between ideas that are central and relevant to the research, and those that are tangential. The Strength parameter is determined by multiplying two sub-parameters. One is selected to map the level of centrality or relevance of the topic to the study, thus its levels were determined according to the areas in the Research Scope diagrams (Figures 1 and 3). The second sub-parameter is the level of importance attributed to the category as set by the researcher, and with reference to explicit power statements made by the interviewees during the interview, such as: "highly important," "necessary condition," and similar. The ***Significance*** parameter is obtained by multiplying the Quantity and Strength parameters and ranked (mapped) using a Quantity-Strength grid, see Figure 4. On this basis, 6 levels of Significance were mapped, as listed in Figure 5.

In the next step, in order to establish the validity and trustworthiness of the study—the degree of correspondence between the data collected and the subject being studied—additional analyses were performed, including crosswise analysis of all the experts' analyses. At this stage, *triangulation* was used to cross-examine the idea categories, and *cross-content analysis* was used to deepen learning, particularly regarding the interrelationships between categories. Finally, as the mapping progressed, categories were combined by

determining primary categories and drawing links between them, in order to focus on the relationships between categories and better understand the full potential inherent in the data gathered. The data collection and analysis at the final stage began with a quantitative view with the *four levels of content analysis*, and proceeded with the mapping created by the cross-content analysis and the triangulation applied.

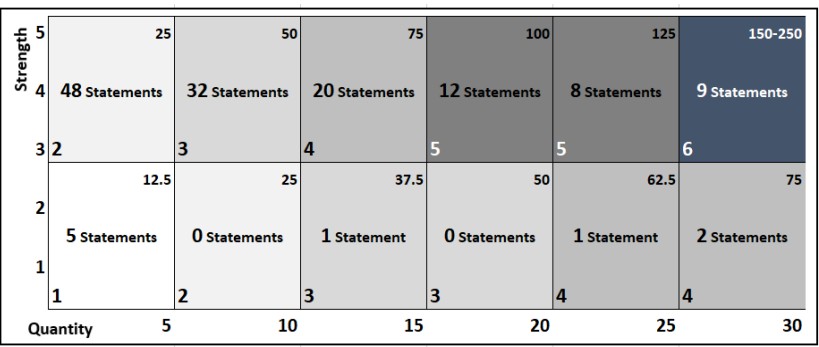

**Figure 4.** Summary of the quantitative results by the levels of Significance using Quantity-Strength grid.

| Summary: | | No. of Statements |
|---|---|---|
| Significance L-1 | Very Low Significance | 5 |
| Significance L-2 | Low Significance | 48 |
| Significance L-3 | Medium Significance | 33 |
| Significance L-4 | High Significance | 23 |
| Significance L-5 | Very High Significance | 20 |
| Significance L-6 | Extremely High Significance | 9 |
| | | **138** |

**Figure 5.** Summary table of the research results by Significance level.

## 4. Results

### 4.1. Quantitative View of the Four Levels Content Analysis

A quantitative view of the results of the four-level content analysis was summarized in a graph of the category distribution using a Quantity-Strength grid showing the number of categories, their Significance rating, and how they were ranked by each expert. The total number of categories is 138. In addition, the Quantity-Strength grid shows the division into areas (segments) of Significance and the number of statements in each area. A summary of this is displayed in Figure 4, based on a graph of the 6 levels of Significance from the Quantity-Strength grid; Figure 5 is a summary table. For instance, the experts rated 20 categories out of the 138 at the 5th level of Significance, meaning each of them rated the statement as having "very high Significance".

### 4.2. Final Results

The final results, after completion of the cross-content analysis and triangulation processes, combined with the analytical (literature) review are summarized in Figure 6. Figure 6 shows the cross-categories found in the cross-analysis stage, according to these parameters: (1) the number of experts whose analysis identified a category which is in the basis of the cross-category; (2) the number of categories included in the basis of the cross-category; (3) the Significance level for each.

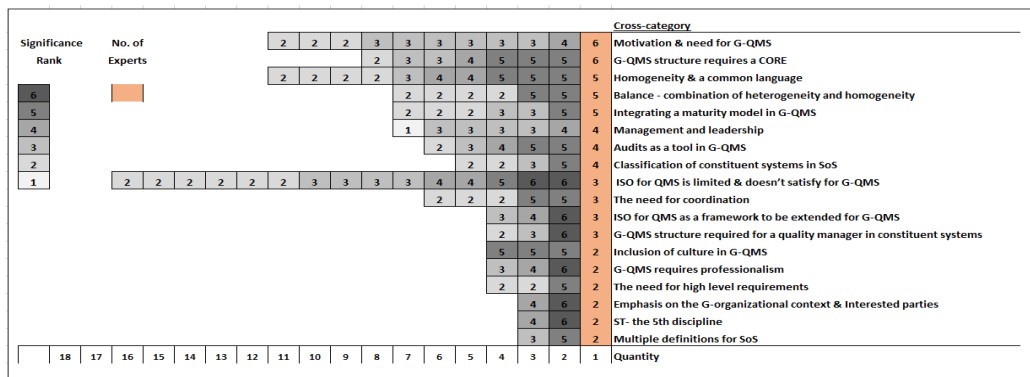

**Figure 6.** Concluding graph of the cross-category analysis according to the no. of experts, quantity and significance.

For instance, the first cross-category "Motivation & need for G-QMS" (1) was mentioned by 6 out of the 7 experts; (2) a total of 10 categories are in the basis of this cross-category (meaning it is in more than one category for some experts); and (3) one ranked it on the 4th level ("High"); six, on the 3rd level; and three, on the 2nd level. It should be noted that although the 2nd level of Significance is labeled "Low Significance", it actually includes categories that were ranked high for Strength (3–5 level), but this was less frequent, as can be seen in the Quantity-Strength grid (Figure 4). This means that they are ranked with a relatively high level of intensity but without much repetition, which reinforces the validity and trustworthiness as to the cross-categories found.

Final results according to the rating shown in Figure 6 can be summarized as follows (from top to bottom):

1.  Motivation and need for G-QMS: G-QMS in global SoS organizations has not been defined and formalized, although there are QMS applications in these organizations, which do include some aspects of globalization. According to all of the experts there is a great need for this. They all stated explicitly that it is very important to develop a G-QMS for global SoS organizations. The terms they used included "necessary condition" and "most important."

2.  G-QMS structure requires a CORE: The idea that the structure of the G-QMS must include a *Cooperate Management Center (CORE)* is very powerful. The CORE of the G-QMS should include a *high level of expertise* in quality management, and have defined *control and reporting channels*. This cornerstone, in the structure of any G-QMS to be defined, was emphasized by all of the experts. Figure 7 illustrates this structure.

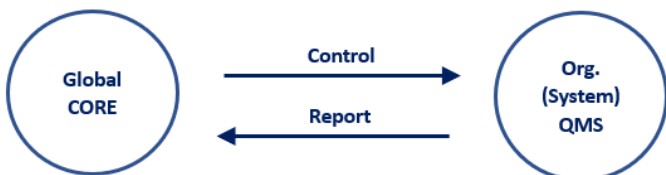

**Figure 7.** A cornerstone in the structure of G-QMS in global SoS organizations.

3.  Homogeneity and a common language: G-QMS must establish a *common language* that includes common terms for the coordination and operation of the G-QMS. For example, setting a known, accepted assessment scale for Quality rates (and thus for the *control* and *reporting channels*) between the G-QMS's CORE and QMSs of all constituent systems organizations. *Homogeneity* and *common language* are the basis for *common processes*. A G-QMS that includes common processes (although not necessarily to all of its processes) will lead to a stronger G-QMS array (common processes are both critical and challenging for a global organization). Other derivatives of this idea include the creation of *basic conditions* for G-QMS, one of which is defining a *set of basic*

*requirements* for each constituent system of the SoS, beyond which concrete extensions can be defined per constituent system.

4. Balance—combination of heterogeneity and homogeneity: Most of the experts frequently and vigorously raised a structural point, namely that an interwoven G-QMS structure is required to *balance* the homogeneity and heterogeneity of the QMSs of the various constituent systems, which they called a key or critical idea. The *balance* between striving for uniformity and preserving heterogeneity must be determined, along with *balancing* the defined function between the G-QMS CORE and the QMS of the constituent system.

5. Integrating a maturity model in G-QMS: The ISO standard model for certification (dichotomous model 0/1) is no longer suitable for G-QMS arrays, and a more extended model is required to support the complexity and scope of G-QMS activity. The idea of incorporating a model that determines maturity levels for an array—e.g., the CMMI, EFQM and similar—was raised directly by most experts. The rating of *maturity model* can focus on the QMSs rating of each of the constituent system in the SoS, as well as rating the global and aggregate level of the G-QMS.

6. Management and leadership: *Management involvement*, *leadership and commitment* are two of the *quality management principles* that form the basis of the ISO standards, which were also mentioned strongly as a condition for effective implementation of the G-QMS.

7. Audits as a tool in G-QMS: The idea that the "ISO standard for QMS as a framework to be extended for G-QMS" is included in point 11, below. However, one of the framework tools—external and internal audits—stood out in its level of intensity and in the number of experts who referred to it. The use of external and internal audits could potentially be very significant for functional expansion, as a balancing tool and a common language for assessment, supervision and control, as well as for exchanging knowledge and information.

8. Classification of constituent systems in SoS: An interesting and important idea that arose concerning the basis of the G-QMS model is the *classification* of the QMSs of the constituent systems in the SoS, with regards to the level of their functional centrality in SoS and their level of complexity. We will elaborate on this idea in sub-Section 5.2.4 of the Discussion. This idea also integrates with others that point to the need for *balance* in the heterogeneous array of G-QMS depending on the nature of the constituent systems of the SoS as well as the nature of the SoS itself.

9. ISO for QMS is limited and doesn't satisfy for G-QMS: The idea that current ISO standards are inadequate for G-QMS in global SoS organizations was stated with great emphasis by three quality management experts, and also received the highest number of repetitions (15, see Figure 6). Indeed, the standards were said to include many limitations regarding the description and requirements needed for G-QMS. One of the limitations raised directly by two experts is that ISO standards do not address the *commonality* (*uniformity*) of the QMS, rather they are self-adapting to any organization. The result is that organizations that are certified to the same ISO standard might actually have different QMSs. The certification model does not facilitate *distinction* as a *basis for comparison* among the QMSs of various organizations. Thus, for example, a G-QMS array based on these standards for the QMSs of the constituent systems of the SoS will not be able to work on this basis alone, and will need additional mechanism/requirements. Two further examples of the limitations of the current ISO standards, which are mentioned in general only. First, the most general requirement to comply with "applicable statutory and regulatory requirements" (clause 4.2 in [2]). This requirement is inadequate even for the QMS of a local organization, and for G-QMS—where the issue of compliance with any law is even more complex it is even less adequate. Second, the requirements set for design and development (clause 8.3 in [2]) are also unsatisfactory, necessitating further reference to global SoS.

10. The need for coordination: *Coordination* is a necessary, *basic condition* for the G-QMS model in global SoS organizations. G-QMS must coordinate with all the QMSs of the constituent systems in the SoS, around the global. This is a particularly important basic condition due to the range of potential G-QMS structures, depending on the nature of the SoS. At minimum there must be coordination between the G-QMS's CORE and the QMSs of each of the constituent systems. This is a basic condition that the system cannot function without. Coordination is also a component in the basis of abovementioned cross-categories (2), (3), (4), and (9).

11. ISO standard for QMS as a framework to be extended for G-QMS: G-QMS can be effectively expanded based on the framework of ISO standards, which could provide very significant added value to any G-QMS model that will be developed. This means that the methods and tools which are considered as groundwork requirements of the ISO standards remain highly relevant in a G-QMS, with the required adjustments.

12. G-QMS structure requirement for a quality manager in constituent systems: Another aspect that relates to the structure required for G-QMS regards the management of QMS in each of the constituent systems in the SoS. *Quality management of the QMS that works on behalf of G-QMS's CORE*, and be built on *professionalism* in the field of quality management, as well as a professional understanding (in terms of technology and practice) of the system.

13. Inclusion of culture in G-QMS: Any G-QMS model must include reference to the *complex organizational culture* that results from globalization. While currently, there is no reference to culture anywhere in ISO standards.

14. G-QMS requires professionalism: G-QMS must be *professional*, with significant knowledge in order to be able to add value to the SoS in its field. Professionalism is especially essential when dealing with highly technologically complex, SoS systems. Quality management is required for the management of a set of processes built in accordance with the product technology, therefore *professionalism* is an important and critical factor.

15. The need for high level requirements: The G-QMS model in global SoS organizations should rely not only on the basic condition of *coordination*, but also include a *set of high-level requirements* that all constituent systems must meet. This creates boundaries, but constituent systems can still "play" within the limits set by the SoS.

16. Emphasis on the G-organizational context and interested parties: A central, critical issue for defining and structuring a G-QMS is how the *context of the organization* and its *interested parties* is addressed. These issues exist in ISO 9001 and are even extended in ISO 9004. However, when it comes to the G-QMS the context of the organizational and interested parties are greater and more complex. Therefore, addressing these issues appropriately will shape the contribution required from G-QMS: defining the scope of the G-QMS array, its boundaries, its environment, as well as the set of G-QMS processes and the interrelationships between them. This idea was raised directly and very seriously by two experts, and also appeared as part of additional statements. For instance, the principle of Hierarchy, one of the main principles in the Systems Thinking discipline, is relevant to this context, and was mentioned twice with Significance level 5—"Very High".

17. Systems Thinking: the 5th discipline: Indeed, the *Systems Thinking perspective* and *principles* can be relevant and contribute to the development of G-QMS in global SoS organizations. They should be used as a *framework* for the development of the G-QMS, and thus contribute their advantages and strengths.

18. Multiple definitions for SoS: SoS has multiple definitions, and there is no consensus about them, as we conclude from the comments of the two SoS experts. This is consistent with the literature that does not yet present a single agreed (official) definition for SoS [19].

The findings provide insights that can serve as a foundation for any definition that might be developed for G-QMS in global SoS organizations, and any model that might be built. Certainly, they should be considered when developing G-QMS in global SoS organizations.

## 5. Discussion: Insights into the Definition, Structure and Model of the G-QMS

The purpose of the study is to create the theoretical foundation of the field of research by using the Grounded Theory concepts and technics combined with the literature review and professional experience. The subjects in the study were mapped and analyzed, and key insights were identified and formulated. The following discussion is based on an analysis of the research topics and the knowledge gleaned from it.

### 5.1. Definitions and Terminology

A definition of Global Organization is a very broad, requiring only that the organization's structure spreads over more than one country, and the development processes and technologies be present in two or more countries. Indeed, each global organization has its own structure. In many sectors this is mostly a consequence of acquisitions and mergers. In general, there varied organizational structures, along a continuum, depending on the organization's changing requirements and goals. Something that was strategic yesterday may already be standard today. The current reality is a multitude of global organizations that have very different, constantly changing structures. All in all, Globalization is a relatively new phenomenon whose meanings are yet to be fully explored. Demeter [44], provides a review of globalization in organizations (with a focus on operations management) and finds that "not all the management research areas are as behind as operations management regarding globalization" and, "organizational implications of globalization are important in practice, but are still under-researched".

SoS also lacks an agreed definition. The current study uses the definition proposed by DoD [45]. However, a more detailed definition will be needed for the further development of G-QMS in global SoS organizations, and should include the attributes and characteristics of the SoS. In Systems Engineering, Systems Thinking is an approach to handling global systems, which considers the components of the system as an ensemble, as a *whole* in a *holistic* way, but also using the principle of *hierarchy*. According to this approach, both SoS and G-QMS should be treated as a *hierarchical system* to which the concept and principles of Systems Thinking can be applied. Therefore, when dealing with the definition and structuring of G-QMS in global SoS organizations, this approach can potentially make a major contribution. Global organizations are also a relatively recent phenomenon, and the study of QMS in global organizations is novel. Although, QMS applications can be found in a global organization, yet G-QMS remains undefined, and without a defined model. It is clear that G-QMS faces unique challenges due to the geographical distribution, and the global element is more complex and significant for management; the conditions of *globalization* are important and influential. At the same time, it is still unclear what contribution *globalization* will make to the G-QMS model.

The current research is focused on G-QMS in global SoS organizations. An analysis of the study findings shows that G-QMS is very significant and a necessary condition for these organizations. G-QMS is one of the managerial issues in the main-management of the SoS, and therefore the management of G-QMS is inseparable from the management of the SoS. As noted, G-QMS in global SoS organizations remain undefined. An agreed definition, defined structures and standards for G-QMS in global SoS organizations are lacking, and there is a high incentive to advance towards their development. Three aspects that should be included in any definition formulated for G-QMS in SoS organizations emerge from the analysis of the current findings. First, reference to the complex *organizational culture* that results from geographical distribution. Second, the requirements of the managerial chapters of ISO standards are the *framework* that can provide significant added value in the basis of any G-QMS that will be defined. Third, the use of the *Systems Thinking perspective* and *principles* as a *framework* should be included when developing G-QMS in global SoS organizations. Subsequently, these components should also be the foundation for developing a model system.

*5.2. Content and Structure (Model) of G-QMS in Global SoS Organizations*

The following analysis of the research findings regarding the model of G-QMS in global SoS organizations addresses the required structure, in relation to both existing ISO standards and our analysis of its specific characteristics and needs, as well as its strengths and potential weaknesses. From these, the base anchors of the model and main factors (or focuses) were mapped. The analysis will be augmented by figures that illustrate the conclusions drawn from the study.

5.2.1. A Cornerstone of the G-QMS Structure

Figure 7 illustrates a cornerstone in the structure of a G-QMS in global SoS organizations. This structure includes the *Corporate Management Center (CORE) of the G-QMS* and the *QMS of the constituent system organization*, which are connected by a *dual channel of control and reporting*. In the list of final results, the cornerstone is rated on the second level of intensity and importance. The G-QMS's CORE includes two additional capabilities. Firstly, it has an umbrella-view over the entire G-QMS system. Secondly, it has high professionalism in the field of quality management. The QMS of the constituent system represents a QMS for local (or more extensive) organization, which is typically certified according to at least one of the international standards for QMS (e.g., ISO 9001). Likely that any G-QMS structure must include this cornerstone. The level of dominance and centralization of the CORE can vary from organization to organization, but a global organization that lacks a CORE entity, meaning it has a completely decentralized G-QMS structure, will have significant weakness, and not be able to provide the function that the SoS requires. Furthermore, the level of dominance and centralization of the same CORE can vary vis-à-vis different constituent systems in the same SoS.

5.2.2. Balance between the G-QMS's CORE and the QMS: A Critical Component

A critical component in the G-QMS structure is the **balance** between the G-QMS's CORE and the QMS of the constituent system. Each of the four components of the model shown in Figure 7 can be defined and constructed across sequence-scales. One scale includes the *balance* between the level of independence of QMS and the level of supervision by the CORE. This scale can be defined by a number of sub-scales, which are also sequence-scales. In this model, these sub-scales are represented by the *two communication channels*. That is, there is a *balance* between the two centers, in both the *Control channel* and the *Reporting channel*. The level of balance in each of the channels varies from one SoS's G-QMS to another, but it is the **balance component** that largely defines the structure of the G-QMS and its character. Figure 8 illustrates this idea, by showing two examples of Systems ("System A" and "System B") with the different *balance location* in their *communication channels*. System A represents a structure with a relatively high level of centralization of the CORE and dependency of the QMS, while System B has a relatively low level of centralization of the CORE and much more independence of the QMS of the constituent system.

5.2.3. G-QMS: Heterogeneous Not Homogeneous

G-QMS is not homogeneous but **heterogeneous**. It consists of several (even many) QMSs with different structures and different maturity levels. Although G-QMS is motivated to create uniformity in language, culture, processes and tools of assessment (see point 3 of the final results), in order to produce a partnership on one hand and an effective managerial mechanism on the other, it must be able to accommodate diversity. This *heterogeneity* stems not only from the aggregate QMSs of the various constituent systems but also from the particular characteristics of the SoS structure. In order for such a G-QMS structure to exist, its base must include the several conditions. First, the QMSs of the constituent systems must be certified according to at least one of the relevant international standards. Different types of standards may be used, including those that incorporate the maturity model, (thereby accommodating QMSs with different levels of maturity). Second, the G-QMS must **coordinate** with each of the QMSs of the constituent systems. Note that there could be a

situation in which a constituent system does not have a QMS that is certified according to an international standard, but is still part of the SoS. In this case, the CORE G-QMS uses *coordination* to determine the threshold requirements for that system. The *coordination factor* as presented in point 10 of the final results is actually the basis of the *two communication channels*, as presented above (and illustrated in Figure 7), and its level is set according to the *balance component* presented above (and illustrated in Figure 8). This means that *coordination* creates the structural fabric of the G-QMS, and through it the interrelationships between the CORE and the QMSs of the constituent systems are defined. The second condition creates the required *coordination* between the CORE G-QMS and each of the other QMSs, and thus leading to *coordination* among the different QMSs. The third base condition for the G-QMS structure is the *coordination* that takes place among the QMSs of the various systems themselves. Figure 9 illustrates the *coordination factor* in the G-QMS structure on *two levels*, as described.

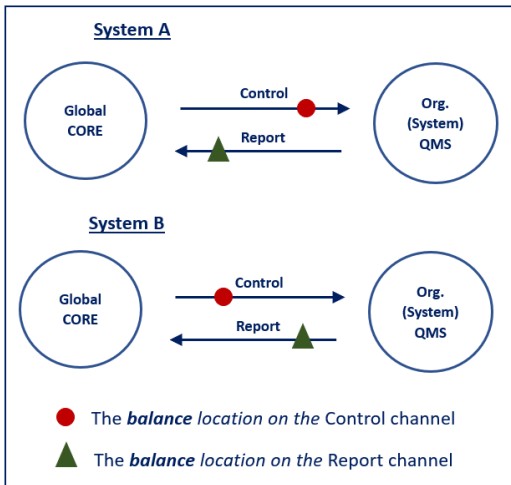

**Figure 8.** The ***balance component*** in the structure of G-QMS in global SoS organizations.

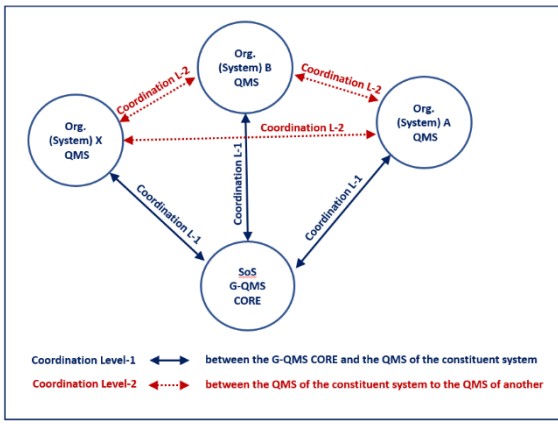

**Figure 9.** The ***coordination factor*** in the structure of G-QMS in global SoS organizations.

5.2.4. Classification of the QMSs of the Constituent Systems

Another dimension to be included when defining a model for G-QMS includes ***classification*** of the QMS of the constituent systems of the SoS. In fact, it's a sorting factor for the constituent systems. The literature documents different structures of SoS, depending on various characteristics [5,16,30] and others, but any given SoS structure includes a set of constituent systems. Each of these systems can be sorted according to a set of characteristics that define the affinity of the systems to the SoS, because some are key systems and others are less central. Some systems are more critical to the SoS than others; some are permanent while others are occasional, etc. QMS should *be classified* according to the *sorting factor* used

for the constituent systems. Accordingly, the structure of the G-QMS array for global SoS organizations is an *aggregate structure* that includes, by definition, a *fabric of QMSs* that differ from each other. This is the **classification dimension**. Consequently, when developing the G-QMS model for global SoS organizations, there is room to define *QMS classes*, according to the different types of constituent systems identified in the SoS.

### 5.2.5. Structures of the ISO Standards as a Framework to Be Extended for G-QMS

Both ISO 9001 and ISO 9004 include requirements and guidelines for QMS, but they lack references to global organizations. In addition, these standards are general and structured for any type of organization, whatever may be, and therefore they lack requirements and guidelines that refer directly to organizations that are complex systems, especially SoS. Thus, the ISO standards are limited and their structures are deficient for defining the structure required for the G-QMS in global SoS organization. However, the structures of the ISO standards do include requirements and guidelines for perspectives and concepts as well as methods and tools which are the **framework** of these standards. This *framework* can give very significant added value to any G-QMS model that would be developed, meaning that the methods and tools that are considered as groundwork requirements of the ISO standards are also highly relevant to G-QMS, with the required adjustments. Therefore, G-QMS can be *effectively expanded*, based on the *framework* of ISO standards. The structures "Figure 2" in ISO 9001 [2] and "Figure 1" in ISO 9004 [3] could be extended in order to include the additional aspects and become the framework for defining to the G-QMS of global SoS organizations. Based on the current study, at least two aspects should be included in this framework: First, reference to the *organizational culture* that is missing in the ISO standards, and second, reference to the *characteristics of the SoS* that are based on the type and nature of the constituent systems. A good source for theses references is Albers et al. [19] who consolidated an exhaustive table of attributes and characteristics for constituent systems and SoS which is based on five additional literature sources.

### 5.2.6. The Expansion of the ISO Standards Framework

The ISO standards framework includes requirements and guidelines that are the managemental methods and tools of the QMS. On that basis, the necessary **expansions** for G-QMS in global SoS organizations should be developed. While sub-Section 5.2.5 presents the idea that the framework of the ISO standards needs to be expanded by including additional aspects relevant to G-QMS in global SoS organizations, this sub-section concerns the idea of *expansions* of the current managemental methods and tools which are already part of the *ISO standards framework*. One example is the Audit tool (clause 9.2 in [2] and clause 10.5 in [3]) concerning the concept and methods for external and internal audits that should be expanded in G-QMS to consider the following, and other, aspects. Is the accreditation body of the G-QMS a single body or does it consist of several accreditation bodies according to the geographical distribution of the organization? What is the internal audits structure? What levels of internal audits are to be defined? Who are the internal auditors to be appointed, and on what levels? How is the internal audits layout to be defined? None of these aspects are included in the ISO standards for QMS, but they are relevant to G-QMS in global SoS organizations and should be developed. Another example, the concept and methods of Management Review (clause 9.3 in [2] and clause 10.7 in [3]) should be expanded in G-QMS to include the defined Review levels for the G-QMS in global SoS organizations, because a single level will be insufficient for G-QMS. The Management Review structure should be defined together with the needed *control and reporting channels*. What are the extra contents to be defined in the Management Review for G-QMS in global SoS organizations, in addition to those which are defined in the ISO? Etc.

### 5.2.7. Systems Thinking Approach as a Framework for G-QMS

The Systems Thinking approach to G-QMS, was rated at the highest level of Significance found in this study, by far, with a score of 295 placing it at the top of scores on the

6th level, "extremely high Significance". This category summarizes the idea of looking at the definition and structure of G-QMS of global SoS organizations according to the principles of Systems Thinking, and *using the perspectives and principles of Systems Thinking as the framework* of a potential model to be developed for G-QMS of global SoS organizations, thereby showing the advantages/strengths of integrating this approach. Among Systems Thinking principles that could be defined in this *framework* are, for instance, the seven Systems Thinking competencies described by Valerdi [46], which can be used in organizational systems and particularly in global systems. Further motivation and support for developing this conclusion is provided by Bashan and Kordova [37], who provide an initial foundation for evaluating and implementing a systems approach to quality management in global organizations by introducing a Systems Thinking framework for a global quality system, by using principles of the Great System Theory (GST). For Example, the *hierarchy principle*, as exemplified in the Results section in point 16 and in the Discussion, Section 5.1, is one of the main Systems Thinking principles (according to the literature). Consequently, the global quality system is characterized by having a *hierarchy* [37].

5.2.8. Dynamic across Sequence-Scales

Any structure that to be developed must be a ***dynamic*** structure across ***sequence-scales***, and according to the type and nature of each SoS organization. Further, a structure can change over-time, depending on the changing nature of the organization, or in response to organizational changes. An example of this kind of *sequence-scale* is the balance between the G-QMS's CORE and the QMS, described in the above sub-clause 5.2.2. In fact, any SoS organization has its own *balance location* between the G-QMS's CORE and the QMSs of the constituent systems, which defines the level of dominance and centralization of the CORE compared to the level of independence of the QMSs. However, this *balance location* can differ for the QMS of each constituent system, in addition to its possible changing over time. Another example could be how the theme of *organizational culture* is developed as part of the model for G-QMS. Organizational culture should also be defined by *sequence-scale*, since it varies from one global SoS organization to another, and also changes over time in each organization. Moreover, G-QMS should be built in a way that supports the organizational strategy, and because organizational strategies now change at an increasingly rapid rate, the structure of the G-QMS must be able to adapt accordingly. The result is a *multitude of structures*, *constantly changing*, and therefore a G-QMS model that can accommodate this is *dynamic*.

*5.3. Model of the G-QMS in Global SoS Organizations*

A graphic depiction of the G-QMS structure based on the principles formulated from the study may be seen in Figure 10. The model includes *CORE of the G-QMS, which has an umbrella-view of the entire G-QMS system and a high level of professionalism in the field of quality management. Situated on the top-level of command and control, it maintains the communication channels, which include a dual channel of control and reporting for each of the QMS in constituent systems.* This is an ***aggregate model*** that *preserves the constituent systems' existing QMS* (even if these are based on different QMS standards). The *aggregate model* is a mosaic that does not strive for uniformity, but rather *is capable of containing heterogeneity*. It includes the *balance component* which defines the level of dominance and centralization of the CORE in relationship to the independence level of the QMSs, and which varies between the QMSs in the same G-QMS. Thus, it defines the model structure which contains a *variety of structural configurations*, as they are obtained from the *balance location*. It also includes the *coordination factor* in the base of the creation of the *structural fabric of the G-QMS*, which is a necessary condition for the realization of the G-QMS structure. In addition, the model defines a *sorting factor* for the constituent systems of the SoS. As a result, the model includes a *classification dimension* for the QMSs of the constituent systems. This special principle adds a dimension to the model for G-QMS in global SoS organizations, one that will, by

definition, enable realization of the *aggregate structure of G-QMS* in a way that will provide the most appropriate and efficient solution, moreover, in accordance with any type of SoS.

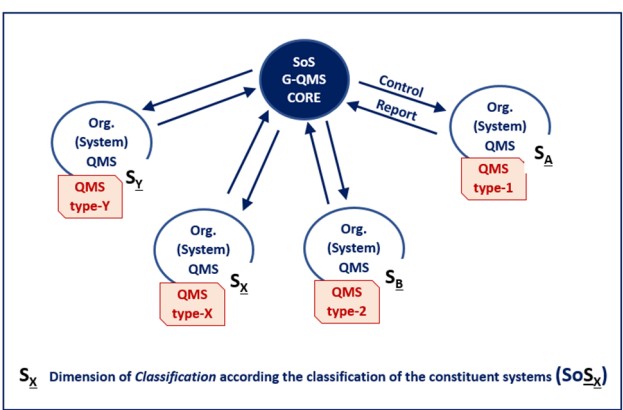

**Figure 10.** Graphic depiction of G-QMS in global SoS organizations.

The model of G-QMS in global SoS organizations shall have a defined framework that outlines the common aspects that are defined at the CORE level and shared by all, while also allowing the QMSs of the constituent systems the freedom to determine their own QMS array and other aspects that the CORE cannot determine for them. The model framework of G-QMS in global SoS organizations shall be developed and extended from the framework of the ISO standards by adding aspects appropriate for a G-QMS in global SoS organizations such as organizational culture and characteristics of the constituent systems of the SoS. Furthermore, current managemental methods and tools included in the frameworks of the ISO standards shall also be expanded. The Systems Thinking perspectives and principles should also be used in the development of the model framework of G-QMS in global SoS organizations. Lastly, the model that will be developed must be adaptable and have a *dynamic structure* that yields a multitude of constantly-changing structures. Thus, the G-QMS model for global SoS organizations must be developed with the ability to contain this *dynamic*.

## 6. Summary and Contributions

The novelty of this study is its integration of several emergent, evolving disciplines in defining a field of research for G-QMS in global SoS organizations, creating a field that is innovative and also extremely relevant for theoretical academic and practical business fields. The research focuses on QMS in today's interconnected, global world that includes a complex, demanding and ever-changing environment and technology, as well as a growing interest in SoS concepts and strategies along with the motivation to extend Systems approaches such as Systems Thinking to these domains. The current study presents a QMS that includes references to global organizational aspects that are missing in existing international standards for QMS, and to various aspects relevant for SoS organizations, which we also found to be deficient and unsatisfactory in the current standards. In order to support these missing aspects, the Process Approach that underlies the ISO international standards for QMS was expanded by introducing Systems approaches, in particular, the Systems Thinking perspectives and principles as well as the potential of expanding the maturity model.

The current research is focused on G-QMS in global SoS organizations, creating a theoretical foundation, and introducing orderly deductions regarding G-QMS, which can be used as base principles for a defining G-QMS and developing a G-QMS model. Analysis of the findings shows that G-QMS is very significant and required for these organizations; it is a necessary condition. G-QMS is a component of the overall management of the SoS, and therefore the management of the G-QMS is inseparable from the management of the SoS itself. Yet, G-QMS in global SoS organizations remains undefined, and lacking

defined structures or standards, leading to significant motivation to advance and develop such systems.

The final results revealed *18 aspects* that should be considered in any definition developed for G-QMS in global SoS organizations, and for any future model. From these, combined with the literature review and professional experience, *8 base anchors* of the model were analyzed and mapped, together with its main factors. Any of the base anchors identified by the study could make its own contribution to any further development in this field. Considering all of them together creates an *initial model* for *G-QMS in global SoS organizations*.

## 6.1. Study Limitations

This paper presents the results of a data analysis based on relativity limited number of experts, although each of the experts was carefully selected based on his seniority and professionalism in the relevant field. Furthermore, each of the experts has his own specific expertise with regards to field of study, and thus not much repetition was included. It should be noted that only one expert for Systems Thinking participated in this study, and this is reflected in the quantification of the results. However, the core of the field of study is covered repeatedly by most of the experts. This limitation challenged the quantitative analysis that underpinned the qualitative analysis, which was performed in depth, and required more understanding and contribution of the professional knowledge and literature background to overcome that.

## 6.2. Directions for Future Studies

The study provides a theoretical foundation for G-QMS in global SoS organizations and the basic foundations and principles for its definition, structures and model. As such, future research should deepen the knowledge of G-QMS in global SoS organizations by establishing this definition and structure. In addition, in accordance with the parameters for further development set in this paper. In these studies, further analysis should focus on the following aspects: First, the attributes and characteristics of the SoS structure, how they are integrated and contribute to the G-QMS structure. Second, deepening the analysis regarding maturity models of all kinds, which received less attention in the present study, and considering also maturity model for the entire G-QMS, not only for the QMSs of the constituent systems. Third, the recommended adaption of the System Thinking perspectives and principles as well as to other Systems approaches.

Another direction for further research and development of the model would be conducting field research on real cases. This direction is particularly relevant, since there is still no formal structure of G-QMS in global SoS organizations, but there are independent applications depending on each and every organization. Therefore, research on real cases is an essential component for developing the body of knowledge.

Further research could review the additional QMS standards, and add aspects from them to any model which is developed for G-QMS in global SoS organizations. In particular, the study of Sectoral QMS Standards should be advanced because they are typically relevant to the domains of SoS. This direction could further contribute to developing a model of G-QMS in global SoS organizations in accordance with the ongoing trend of differentiation in the QMS standardization, and promote models which are tailor-made for specific domains.

**Author Contributions:** Conceptualization, N.A., S.K. and S.S.; methodology, N.A., S.K. and S.S.; validation, N.A.; formal analysis, N.A.; investigation, N.A.; resources, N.A.; data curation, N.A.; writing—original draft preparation, N.A., S.K. and S.S.; writing—review and editing, N.A., S.K. and S.S. All authors have read and agreed to the published version of the manuscript.

**Funding:** This research received no external funding.

**Conflicts of Interest:** The authors declare no conflict of interest.

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
