# Peer review of "Global Quality Management System (G-QMS) in Systems of Systems (SoS)—Aspects of Definition, Structure and Model"

_systems, doi:10.3390/systems10040099_

Round 1

Reviewer 1 Report

In the Rationale subsection, there are repeated phrases from the abstract. I suggest summarising the text of the abstract as to avoid such repetition.

I suggest enriching the Literature review with references to previous studies.

Author Response

Referee #1

Comments and Suggestions

Response

1

In the Rationale subsection, there are repeated phrases from the abstract. I suggest summarizing the text of the abstract as to avoid such repetition

We thank Referee #1 for the suggestion and revised the abstract accordingly.

2

I suggest enriching the Literature review with references to previous studies.

We thank Referee #1 for the enriching suggestion. The literature review that we conducted, includes some previous studies that can be referenced to this new field of study. However, in response to the referee suggestion we reevaluated the entire Literature in the base of this study and, indeed, there are not much since this field of study novelty combined the four disciplines to a coherent field of study.

Reviewer 2 Report

Dear authors

I send you my congratulations, for your interesting article, as well as for approaching the subject from a less frequent methodological perspective, but very rich in contributions. On the other hand, it is carefully elaborated in its structure and publication standards of excellent quality.

Kind regards

Author Response

Referee #2

Comments and Suggestions

Response

1

I send you my congratulations, for your interesting article, as well as for approaching the subject from a less frequent methodological perspective, but very rich in contributions. On the other hand, it is carefully elaborated in its structure and publication standards of excellent quality.

We thank Reviewer #2 for his/her congratulations! We appreciate his/her encouraging review feedback.

Reviewer 3 Report

The theme is relevant and innovative in analyzing and discussing this set of concepts. Introducing the concept of globalization in QMS and SoS is an advance in organizational management theory. The relationship of the concepts makes sense, although it is not the peaceful boundary line between them.

In the Abstract, the references "Methodology:" and "Results and conclusions:" must be plain text, as in "The purpose...".

The methodology and research design are appropriate but limited in the number of experts involved (7), undermining the reliability of the results.

The Reseach Design states that "The analytical component included both the content analysis methods of the research data and secondary sources (literature)". However, it is not clear which were the sources and the criteria adopted for the selection of sources.
This is an exploratory study with the identification of a relevant line of research. The study adds value by creating a starting theoretical framework.

Author Response

Referee #3

Comments and Suggestions

Response

1

The theme is relevant and innovative in analyzing and discussing this set of concepts. Introducing the concept of globalization in QMS and SoS is an advance in organizational management theory. The relationship of the concepts makes sense, although it is not the peaceful boundary line between them.

We thank Referee #3 for his/her positive comments that finds the topic of this paper relevant and innovative, and with an advance in organizational management theory. Indeed, since this field of study is innovative in its concepts by integration of several novel and quickly developing disciplines, there is a room to refine their boundaries in further researches. 

2

In the Abstract, the references "Methodology:" and "Results and conclusions:" must be plain text, as in "The purpose...".

We thank Referee #3 for his/her comment and revised that in the Abstract accordingly.

3

The methodology and research design are appropriate but limited in the number of experts involved (7), undermining the reliability of the results.

This is an excellent point made by Referee #3. Indeed, the limited number of experts is the main limitation of this study.  This point is inscribed and explained in paragraph 6.1.

Also, as it is written on at the beginning of paragraph 3.2.1., the experts for the study were carefully selected and each of them is considered “state of the art” in his/her respective field. We believe they represent many other experts in their respective field. In future we may extend and verify the finding described in this paper with additional experts.  

4

The Research Design states that "The analytical component included both the content analysis methods of the research data and secondary sources (literature)". However, it is not clear which were the sources and the criteria adopted for the selection of sources.

We thank Referee #3 for this delicate notice. This is precisely the Research Design. Here are only some examples, from the text of the manuscript: in paragraph 4.2 – Final Results: (1) in result 5 are shown for the standards CMMI and EFQM as example for standards that include maturity model. (2) in result 6 we refer to the main quality management principles in the base of the ISO standards. (3) in result 7, we refer to the Audit tool of the ISO standards, and then in the Discussion in paragraph 5.2.6 we give the references “clause 9.2 in [2] and clause 10.5 in [3]”. Also, in this paragraph, we add the references “clause 9.3 in [2] and clause 10.7 in [3]”. (4) result 9 refers to clause 4.2 in [2] and clause 8.3 in [2]. (5) result 18 refers to [19]. (6) in in the Discussion in paragraph 5.1 we refer to [44]. (7) in the Discussion in paragraph 5.2.4 we refer to [5], [16] and [30]. (8) in the Discussion in paragraph 5.2.5 we are based on the structures “Figure 2” in ISO 9001 [2] and ‘Figure 1” in ISO 9004 [3]. Also, in the paragraph we refer to [19]. (9) Another example is given in the Discussion in paragraph 5.2.7, when first, we give the direct score of the parameter Significance with the score of 295 as obtained from the data analysis and also, we refer to [46] and [37] and then we do an additional reference to [37].

5

This is an exploratory study with the identification of a relevant line of research. The study adds value by creating a starting theoretical framework.

We thank again Referee #3 for finding our study with the identification of a relevant line of research, and a study that adds value by creating a starting theoretical framework. The reviewer is right that this is a starting theoretical framework, and we plan to continue this research.

Reviewer 4 Report

The authors raised a very complex problem in the manuscript. The manuscript shows the link between Quality Management System, Global Quality Management System, System of Systems, Systems theory and Systems thinking in a clear and interesting way. It is characterized by a logical structure and presents the intentions of the authors in an exhaustive way.

Author Response

Referee #4

Comments and Suggestions

Response

1

The authors raised a very complex problem in the manuscript. The manuscript shows the link between Quality Management System, Global Quality Management System, System of Systems, Systems theory and Systems thinking in a clear and interesting way. It is characterized by a logical structure and presents the intentions of the authors in an exhaustive way.

We thank Referee #4 for his/her positive and supportive comments. Indeed, we elaborate a very complex field in the manuscript, that links between Quality Management System, Global Quality Management System, System of Systems, Systems theory and Systems thinking. While doing so with much attention to the methodological and structure in our work. We highly appreciate these comments.

Reviewer 5 Report

The article addresses a very interesting topic, applies an appropriate methodology and presents the results clearly. The purpose of the study is described in a logical, comprehensible, and explicit manner. A notable argumentation in support of research pointing to gaps in the literature is established. The significance of the research is clearly established. Also, the authors presented the limitations of the research and highlighted the future directions of this research.

However, the author (s) should also consider the following suggestions:

-For all the figure presented in the paper, the author (s) should mention the source.

3.2. Research Design

- Research Design section should be improved with information about the research period. When were the interviews with the experts conducted?

5. Summary and Contributions

-Who might benefit from the results of this study? The author (s) should better highlight the implications of the findings on all stakeholders (organizations, researchers). Also, perhaps the contributions (implications of this study) should be grouped into theoretical and practical contributions.

- Line 900 and Line 1045: There is a space/word marked in yellow.

References

-For some sources in the list of references the DOI is mentioned in a different way (for example reference no. 28 versus reference no.29). I would suggest to the author to write the DOI in the same way in the entire references list.

Author Response

Referee #5

Comments and Suggestions

Response

1

The article addresses a very interesting topic, applies an appropriate methodology and presents the results clearly.

We thank Referee #5 for his/her positive and supportive comments. We were pleased to read that the reviewer finds our manuscript includes appropriate methodology and presentation of results in a clear way, as well as to the purpose of the study, literature review, and the attention which is given to the study limitations and future directions. We highly appreciate these comments.

2

The purpose of the study is described in a logical, comprehensible, and explicit manner.

3

A notable argumentation in support of research pointing to gaps in the literature is established.

4

The significance of the research is clearly established.

5

Also, the authors presented the limitations of the research and highlighted the future directions of this research.

6

For all the figure presented in the paper, the author (s) should mention the source.

We thank Referee #5 for the right and important comment. In this manuscript all the presented figures are original and were not referenced from others source. Therefore, there is no source to be mentioned.

7

3.2. Research Design:

Research Design section should be improved with information about the research period. When were the interviews with the experts conducted?

We thank Referee #5 for this comment that clarify the research period.

Accordantly, we added “The study interviews were conducted in a planned sequence during a year period” in paragraph 3.2.1 - Exploratory Study.

8

5. Summary and Contributions:

Who might benefit from the results of this study? The author (s) should better highlight the implications of the findings on all stakeholders (organizations, researchers). Also, perhaps the contributions (implications of this study) should be grouped into theoretical and practical contributions.

We thank Referee #5 for this comment that improve to clarify the implications of the study findings. Indeed, in paragraph 6 - Summary and contributions - is written that the researchers would be benefited from the study contributions, by using the term "theoretical academic" and organizations by using the term "practical business fields".

9

Line 900 and Line 1045: There is a space/word marked in yellow.

We thank Referee #5 for his/her sharp attentions and did the corrections accordingly. 

10

References:

For some sources in the list of references the DOI is mentioned in a different way (for example reference no. 28 versus reference no.29). I would suggest to the author to write the DOI in the same way in the entire references list.

We thank Referee #5 for his/her sharp attentions and did the corrections accordantly. 

In addition, we went over the references list again, made corrections, to ensure a uniform and accurate record.
